# CD63-Mediated SARS-CoV-2 RBD Fusion Neoantigen DNA Vaccine Enhances Antitumor Immune Response in a Mouse Panc02 Model via EV-Targeted Delivery

**DOI:** 10.3390/vaccines13090977

**Published:** 2025-09-16

**Authors:** Guang Liu, Ziqing Yuan, Ziyi Wu, Qiyv Yang, Tingbo Ding, Ker Yu, Jibin Dong

**Affiliations:** 1Department of Pharmacology, School of Pharmaceutical Sciences, Fudan University, Shanghai 200120, China; 22211030018@m.fudan.edu.cn (G.L.); 23211030041@m.fudan.edu.cn (Z.Y.); 24211030095@m.fudan.edu.cn (Z.W.); 24211030102@m.fudan.edu.cn (Q.Y.); 2Experiment & Teaching Center, School of Pharmaceutical Sciences, Fudan University, Shanghai 200120, China; tbding@shmu.edu.cn

**Keywords:** extracellular vesicles, CD63, dendritic cell, DNA neoantigen vaccine

## Abstract

Background: Although DNA vaccines offer a flexible platform for tumor immunotherapy, their weak immunogenicity remains a key limitation. This study aimed to improve the immunogenicity of DNA vaccines by enhancing the efficiency of tumor neoantigen delivery through extracellular vesicles (EVs), thereby promoting stronger dendritic cell (DC) activation and antitumor responses. Methods: A novel DNA vaccine (pCSP) was engineered by fusing tumor-specific neoantigens to the EV-associated protein CD63 and incorporating a SARS-CoV-2 receptor-binding domain (RBD) fragment to facilitate EV uptake by DCs. The resulting EVs were expected to carry neoantigens into the immunoproteasome for major histocompatibility complex I (MHC-I) presentation. The immunological and antitumor effects of pCSP were assessed through in vitro functional assays and in vivo experiments in a murine pancreatic cancer model. Safety was evaluated through histological and biochemical analyses. Results: In vitro, pCSP significantly promoted EV internalization by DCs by approximately twofold and enhanced their immune activation, as evidenced by elevated cytokine production. In vivo, pCSP markedly suppressed tumor growth with a decrease in volume by over 70% relative to controls, boosted CD8+ T cell responses, and increased immune infiltration into the tumor microenvironment. Safety assessments revealed that while liver/kidney function markers were within physiological ranges, mild inflammatory infiltrates were consistently observed in the lungs, indicating a localized safety concern that warrants further monitoring. Conclusions: The pCSP vaccine enhances the immunogenicity of neoantigen DNA vaccines by improving EV uptake and immune activation in DCs. These findings provide a potential strategy for improving DNA vaccine efficacy in the context of cancer immunotherapy while maintaining acceptable safety.

## 1. Introduction

Therapeutic DNA cancer vaccines are considered a promising strategy for activating the immune system to fight cancer [1]. Preclinical and clinical trials of plasmid DNA-based vaccines have demonstrated a good safety profile and ability to effectively activate broad and specific immune responses [2,3]. However, early human clinical trials have shown that the immunogenicity of DNA vaccines is far below what is expected [4]. To enhance the immune effects of DNA vaccines, two major strategies have emerged from the analysis of past preclinical and clinical trial results [5]. First, the immunogenicity of plasmid DNA vaccines can be improved by selecting and optimizing the inserted antigens. Specifically, with the rapid development of next-generation sequencing, personalized immunotherapy based on mutation-associated neoantigens (MANA) from individual cancer patients has gained considerable attention [6]. MANA is caused by tumor genome mutations and is presented by the major histocompatibility complex (MHC) on the cell surface. Due to the lack of these mutated antigens in normal tissues, they typically do not trigger immune system tolerance and, instead, more effectively activate antitumor immune responses. Considering the heterogeneity of tumors, very few neoantigens are shared among cancer patients, and most tumor neoantigens are passenger mutations rather than driver mutations. This necessitates the development of personalized neoantigen vaccines. Additionally, multiple immune epitopes are often linked together in neoantigen vaccines to overcome issues such as the insufficient immunogenicity of single epitopes and antigen mutation or loss in tumor cells [7,8,9]. Several personalized therapeutic cancer vaccines targeting MANA have successfully induced tumor-specific T cell responses and blocked tumor growth in preclinical and early clinical trials [7,10,11,12]. These results suggest the potential of multi-epitope personalized DNA cancer neoantigen vaccines. The second strategy involves combining DNA vaccines with other adjunctive therapies, which can enhance vaccine activity by weakening immune suppression in the tumor microenvironment or increasing the activity and number of immune cells [13]. Additionally, strategies to enhance the immunogenicity of DNA vaccines include improving vaccination protocols [14], modifying delivery routes, and adding adjuvants [15]. Moreover, the use of EVs to increase antigen presentation has been explored [16].

In past studies, researchers developed DNA vaccines encoding antigens (Ags) fused to CD63, enabling the delivery of Ags into EVs. These vaccines effectively induced Ag-specific T-cell responses—particularly from CD8+ T cells—in mouse tumor models, significantly suppressing tumor growth [16]. CD63 is a tetraspanin protein widely found on the surface of EVs, characterized by two extracellular loop domains [17]. Another study engineered a pH-sensitive tetraspanin (TSPAN) reporter gene based on CD63’s extracellular structure, in which the first extracellular loop of CD63 was fused with a pH-sensitive GFP (pHluorin) to visualize the fusion of endosomes with the plasma membrane [18,19]. Inspired by these findings, we hypothesized that inserting a DC-targeting domain into the extracellular structure of CD63 in DNA vaccines could allow EVs loaded with antigen proteins to target DCs.

With the ongoing spread of the COVID-19 pandemic, SARS-CoV-2 has become widely recognized. Several receptors—including ACE2, NRP1, and L-SIGN—have been shown to bind the receptor-binding domain (RBD) of SARS-CoV-2. In addition, the dendritic cell-specific intercellular adhesion molecule-3-grabbing non-integrin (DC-SIGN) has been recognized as playing a functional role in COVID-19 and has been shown to interact with the SARS-CoV-2 RBD [20,21]. DC-SIGN is a dendritic cell-specific C-type lectin that serves as a cell adhesion receptor mediating DC migration and T cell activation, which also has the ability to internalize and present Ags. Data suggest that the adhesive receptor DC-SIGN can be targeted to induce antitumor immunity by specifically targeting tumor antigens to DCs [22]. Given this, we hypothesize that adding the SARS-CoV-2 receptor-binding domain (RBD) to the extracellular domain of CD63 can facilitate the delivery of EVs to DCs. This approach leverages the interaction between the SARS-CoV-2 RBD and DC-SIGN, which may enhance the uptake of EVs by DCs, improving antigen presentation and potentially augmenting immune responses.

Some highly mutated cancers, such as melanoma, may be sensitive to monotherapy with immune checkpoint inhibitors due to their high neoantigen load and abundant tumor-infiltrating lymphocytes (TILs). In contrast, pancreatic ductal adenocarcinoma (PDAC) is a tumor with low immunogenicity, fewer predicted neoantigens, lower immune infiltration, and poor response rates to immune checkpoint inhibitors [23]. Therefore, to enhance the immunogenicity of DNA vaccines for PDAC, this study targets neoantigens from the pancreatic cancer cell line Panc02 in C57BL/6 mice. Using whole-exome sequencing (WES) data, RNA sequencing (RNASeq) data, and the computational immunogenicity prediction algorithm pVACtools, several neoantigen epitopes that can be presented by mouse MHC I molecules were selected. These epitopes were then linked to the C-terminal of CD63, and a SARS-CoV-2 RBD fragment was inserted into the small extracellular loops of the CD63 protein to create a novel multi-epitope DNA vaccine fusion protein, CSP. The vaccine uses CD63 to bring antigen fragments into EVs secreted by the cells, and the SARS-CoV-2 RBD fragment targets DCs. The activated DCs then present the antigens to cytotoxic T cells, which next migrate to the tumor site, recognize mutated tumor neoantigen peptides presented by internally expressed MHC I molecules, and kill the tumor cells (Figure 1). We characterized and validated the targeting ability of CSP-modified EVs in vitro, followed by in vivo testing in a tumor-bearing mouse model to evaluate the vaccine’s effect on cellular immune responses and tumor growth. This study provides insights into the development of safe and highly immunogenic therapeutic DNA cancer vaccines.

## 2. Materials and Methods

### 2.1. Somatic Mutation Analysis and Screening

The sequencing datasets were sourced from the NCBI SRA database (https://www.ncbi.nlm.nih.gov/sra), including the Panc02 WES dataset SRR14239937 [24], the Panc02 RNAseq dataset SRR10112009 [25], and the C57BL/6 mouse blood WES dataset SRR11127467 [26], which served as the normal controls. The datasets were initially used to identify neoantigens using OpenVax [27]. Briefly, the tumor and normal WES FASTQ files were aligned with the mouse mm10 genome using the BWA MEM algorithm. The aligned tumor and normal exome reads were processed through several steps using the Genome Analysis Toolkit 3.7 (GATK 3.7): marking duplicates, inserting alignment results, and recalibrating the genome quality. Somatic variant calling was performed by running Mutect 2 and Strelka 1. The RNA-seq data from the tumor samples were aligned with the same reference genome using STAR. The aligned RNA-seq reads were divided into two groups: those spanning introns and those fully located in exons (determined by CIGAR). The latter group was passed through the indel algorithm, and the two groups of reads were merged. Next, the merged Mutect and Strelka variant calling format (VCF) files were further modified to meet the input file requirements for the next steps. The updated VCF files were then processed using the pVACtools Docker version and the pVACseq program [28,29]. Finally, pVACvector was used to assemble the mutant peptide sequences. These were then processed with a codon optimization tool to generate the corresponding DNA sequences for the tumor neoantigens, enabling efficient translation in mice.

### 2.2. DNA Vaccine Construction

The DNA sequence encoding the tumor neoantigens was ligated to the C-terminal of the full-length CD63 DNA sequence [30] from C57BL/6 mice. As a linker, a four-amino acid HHHD corresponding to a DNA sequence was inserted between the CD63 and the neoantigen gene. The SARS-CoV-2 RBD-associated sequence was then inserted into the small extracellular loops of the CD63 protein to mimic the study design from the pHluorin research [18]. The constructed DNA was then introduced into the pcDNA3.1+ mammalian expression vector, creating the plasmid pCSP, which encodes the fusion protein CSP. Five control plasmids were also designed: plasmid pC containing only the CD63 sequence, plasmid pP containing only the neoantigen sequence, plasmid pCS containing both the SARS-CoV-2 RBD sequence and the CD63 combination sequence, plasmid pCP containing the CD63 and neoantigen combination sequence but without the SARS-CoV-2 RBD sequence, and an empty plasmid serving as the negative control (Figure 2B). All plasmids were synthesized by SYNBIO Technologies Co., Ltd. (Suzhou, China). The plasmids were amplified in Escherichia coli DH5α cells and the plasmid DNA was purified using an Endofree Maxi Plasmid Kit (DP117, TIANGEN Biotech Co., Ltd., Beijing, China), according to the manufacturer’s protocol. The recombinant plasmid was then confirmed via double digestion with the restriction endonucleases XbaI and BamHI (TaKaRa, Dalian, China), and the nucleotide sequence of the recombinant plasmid was verified through DNA sequencing.

### 2.3. Protein Structure Prediction

The FASTA file corresponding to the amino acid sequence of the fusion protein was submitted to AlphaFold2 [30] to predict the 3D structure of CSP. The protein structure was then visualized and analyzed using the PyMOL software v3.1. The amino acid sequence of the fusion protein was also submitted to DeepTMHMM (https://dtu.biolib.com/DeepTMHMM/) to predict the transmembrane structure of the protein [31].

### 2.4. Cell Culture and Preparation

HEK293T cells (Human embryonic kidney 293T cells, RRID: CVCL_0063) were purchased from the Cell Bank of the Chinese Academy of Sciences, panc02 cells (C57BL/6 mouse pancreatic cancer cells, RRID: CVCL_D627) were obtained from Meixuan Biotechnology Co., Ltd. (Shanghai, China), and DC2.4 cells (C57BL/6 mouse bone marrow-derived dendritic cells, RRID: CVCL_J409) were kindly provided by Wuhan Pricella Biotechnology Co., Ltd., Wuhan, China. HEK293T and Panc02 cells were cultured in Dulbecco’s Modified Eagle Medium (DMEM, Meilumbio, Shanghai, China) supplemented with 10% fetal bovine serum (FBS, ExCell, Shanghai, China), 100 U/mL penicillin, and 100 U/mL streptomycin (Meilumbio, Shanghai, China). DC2.4 cells were cultured in RPMI-1640 medium (Gibco, Grand Island, NY, USA) supplemented with 10% FBS, 100 U/mL penicillin, and 100 U/mL streptomycin. All cells were maintained in a 37 °C, 5% CO_2_ incubator, and subculturing was performed when the cell density reached 80–90%.

### 2.5. EV Isolation and Characterization

When HEK293T cells reached approximately 80% confluence, various DNA constructs were transiently transfected into the cells using Lipofectamine 3000 (Life Technologies, Gaithersburg, MD, USA) following the manufacturer’s protocol. The cells were then cultured for 48 h in serum-free medium specifically designed for exosomes (Yeasen, Shanghai, China). Cell culture supernatants were collected and centrifuged at 3000× *g* for 10 min. EVs were isolated from the cell culture medium using an Exosome Isolation Kit (Yeasen, Shanghai, China), according to the manufacturer’s instructions. Briefly, 2.5 mL of total exosome isolation reagent was added to 10 mL of cell culture supernatant, incubated overnight at 4 °C, and then centrifuged at 10,000× *g* for 1 h at 4 °C. The EV pellet was resuspended in PBS. To determine the concentration and hydrodynamic diameter of the EVs, the isolated EVs were diluted and analyzed using a nanoparticle tracking analyzer (ZetaView Particle Metrix, Inning am Ammersee, Germany). The morphology of the EVs was observed under a transmission electron microscope (Hitachi TEM HT7700, Tokyo, Japan). The expression of related proteins was assessed using Western blot analysis, as described below.

### 2.6. Western Blotting

Forty-eight hours after HEK293T cells were transfected, the transfected cells were harvested, and the culture medium containing the transfected cells was centrifuged at 2000× *g* for 30 min. The cells were lysed using RIPA lysis buffer (with PMSF) (Solarbio, Beijing, China) and incubated on ice for 15 min. The lysates were then centrifuged at 12,000× *g* for 15 min at 4 °C. Total protein was collected from the cells, and EV samples were isolated from the supernatant. Protein concentration was determined using the BCA Protein Assay Kit (Beyotime, Shanghai, China). EVs and cell lysates were reduced with SDS-PAGE sample buffer (Beyotime, Shanghai, China) and heated at 95 °C for 5 min or 37 °C for 30 min. Equal amounts of samples were subjected to 10% sodium dodecyl sulfate polyacrylamide gel electrophoresis (SDS-PAGE) and transferred to a PVDF membrane (Immobilon, MilliporeSigma, Darmstadt, Germany). The membrane was blocked at room temperature for 2 h with 5% BSA (Solarbio, Beijing, China) and then incubated overnight at 4 °C with the following primary antibodies: anti-CD63 mAb (1:5000, A19023, Abclonal, Shanghai, China), anti-SARS-CoV-2 Spike RBD pAb (1:4000, A20135, Abclonal, Shanghai, China), anti-TSG101 mAb (1:5000, A5789, Abclonal, Shanghai, China), anti-CD81 mAb (1:5000, A4863, Abclonal, Shanghai, China), and anti-Calnexin mAb (1:5000, A4846, Abclonal, Shanghai, China). After three washes with Tris-buffered saline containing Tween 20 (TBST, Servicebio, Shanghai, China), the membrane was incubated at room temperature for 2 h with HRP-conjugated goat anti-rabbit secondary antibody (1:5000, HA1001, Huabio, Hangzhou, China). After three additional washes with TBST, the membrane was treated with chemiluminescent HRP substrate ECL reagent (Meilunbio, Shanghai, China), and the bands were visualized and recorded using the ChemiDoc MP chemiluminescence gel imaging system (BIO-RAD, Hercules, USA).

### 2.7. RNA Isolation and Quantitative Real-Time PCR

HEK293T cells were transfected with each DNA construct and cultured for 48 h. The cells were washed twice with PBS, and total RNA was extracted using Trizol (Novizan, Nanjing, China). According to the manufacturer’s instructions, 1 μg of mRNA was reverse transcribed into cDNA using the Hifair^®^ II 1st Strand cDNA Synthesis SuperMix (Yeasen, Shanghai, China). Real-time quantitative PCR was performed using the 2× Universal SYBR Green Fast qPCR Mix (RK21203, ABclonal, Wuhan, China). Reactions were conducted and analyzed using the CFX Connect™ Fluorescent Quantitative PCR Detection System (BIO-RAD, Hercules, CA, USA). Gene expression levels were normalized to GAPDH expression, and relative gene expression was calculated using the 2−ΔΔCq method. The primer sequences were as follows: mouse CD63 forward, 5′-CAGAGTCCCCGATTCTTGCT-3′, reverse, 5′-TGCTATAGTCTCCACGCAGC-3′; RBD forward, 5′-CATTGCAGACACAACCGACG-3′, reverse, 5′-GCACCCACTGCAATGATGAC-3′; Antigen forward, 5′-GTGCCTAGACCCCACAACTC-3′, reverse, 5′-AGGTACACACTGTCGTGCAG-3′.

### 2.8. DC2.4 Cell Uptake of EVs

To study the cellular uptake of EVs, DC2.4 cells were seeded in a 24-well plate at a density of 5.0 × 10^4^ cells per well and incubated overnight. When 80–90% confluency was reached, EVs labeled with PKH76 fluorescent dye (UR52303, Umibio, Shanghai, China) were incubated with the DC2.4 cells at 37 °C for 3 h [32]. The cells were then washed three times with PBS and harvested. Cell uptake was detected using flow cytometry (SA3800, Sony, Tokyo, Japan), and fluorescence intensity was analyzed using the FlowJo software v10 (TreeStar, San Carlos, CA, USA).

### 2.9. DC2.4 Cell Cytokine Release

DC2.4 cells were seeded in a 96-well plate at a density of 1 × 10^4^ cells per well and incubated overnight before treatment. EVs extracted from cell culture supernatants of various DNA constructs were diluted in Opti-MEM to a final volume of 0.1 mL and added to each well. DC2.4 cells treated with LPS or Opti-MEM were used as positive and negative controls, respectively. The cells were incubated at 37 °C for 8 h, and the supernatants were then collected. The levels of TNF-α and IL-6 in the supernatants were measured by enzyme-linked immunosorbent assay (ELISA) according to the manufacturer’s instructions (Excell, Shanghai, China).

### 2.10. Spleen Cell Isolation and Cytokine Analysis

Plasmids pCS, pP, pCP, and pCSP (100 μg) or an empty plasmid were intramuscularly (i.m.) injected into the tibialis anterior muscle of C57BL/6 mice. Mice were immunized three times with a 7-day interval. Seven days after the final immunization, the mice were euthanized, and their spleen lymphocytes were isolated under sterile conditions to be prepared into single-cell suspensions. A total of 1 × 10^7^ spleen lymphocytes were cultured in complete medium containing 1 μg of each tumor neoantigen peptide (peptide pool: ISVTESSL, MAAAFMGM, KALRTMTAI, VMVVNTWQV, STPKRYTVL, AALGSPAPV, SSLNTLSSL, TSKIYPIL, VSPTFKEF, NSISYTLL, VKIENGNEI, LAINNIRLI, and VSVENEASA) for 12 h. All peptides were synthesized by Sangon Biotech Co., Ltd. (Shanghai, China) with a purity of over 95%. The supernatants were harvested, and the concentrations of IFN-γ, TNF-α, IL-2, and IL-6 in the cell supernatants were determined by ELISA according to the manufacturer’s instructions (Excell, Shanghai, China).

### 2.11. In Vitro Cytotoxicity Assay

To assess antigen-specific cytotoxic T lymphocytes (CTLs), Panc02 cells were first labeled with 5 μM 5,6-carboxyfluorescein diacetate succinimidyl ester (CFSE, Beyotime, Shanghai, China) for 8 min and then co-cultured with activated spleen cells from mice at three different effector-to-target ratios (E/T ratios were 50:1, 25:1, and 12.5:1) at 37 °C for 24 h. After incubation, adherent cells were collected and stained with propidium iodide (PI, Servicebio, Shanghai, China) in the dark at room temperature for 15 min. The proportion of CFSE+/PI+ cells among CFSE+ cells, determined using flow cytometry, represents the percentage of tumor cells killed by the immune-modulated spleen cells after activation.

### 2.12. In Vivo Tumor Treatment Strategy

C57BL/6 mice (female, 5–6 weeks old) were purchased from SLAC ANIMAL Co., Ltd. (Shanghai, China) and were acclimated for at least one week in the Fudan University School of Pharmacy Animal Center under standard conditions of 23 ± 2 °C and 50% ± 10% relative humidity, before being included in the study. On day 0, after being anesthetized with 3% isoflurane inhalation (R510-22-10, RWD Life Science Co., Ltd., China), 2 × 10^5^ Panc02 cells were subcutaneously (s.c.) injected into the right lower flank of the mice. Three days after tumor inoculation, the mice were randomly divided into 5 groups, with 6 mice in each group. On days 4, 11, and 18 after tumor inoculation, each mouse was injected with 100 μg of the DNA vaccine (pCS, pP, pCP, pCSP, or an empty plasmid) in a total volume of 50 μL into the tibialis anterior muscle of the right hind limb. The DNA vaccines were dissolved in physiological saline. When the tumor size exceeds 2 cm^3^, the mice are euthanized. On day 39, the mice were euthanized, and tumor tissues and spleens were dissected for further evaluation. Specifically, the mice were placed in an anesthesia chamber ventilated with 3% isoflurane until they were fully unconscious. Subsequently, the animals were euthanized by cervical dislocation, after which their organs were collected for further analysis (Figure 5A). Every week, the body weight and tumor volume of tumor-bearing mice were measured two to three times using calipers. Tumor volume was calculated using the following formula: Tumor volume = (length × width^2^)/2 (mm^3^).

### 2.13. RNAseq

After euthanizing the mice in the treatment model, tumors were quickly dissected into small pieces and transferred to RNAlater (Solarbio, Beijing, China) and placed at 4 °C overnight. Total RNA was extracted using TransZol up (TransGen Biotech, Beijing, China). After RNA extraction, RNA integrity was assessed using 1.5% agarose gel electrophoresis or a fragment analyzer, and RNA concentration and purity were determined using a Nanodrop. mRNA was enriched from the samples using magnetic beads with Oligo (dT). Transcriptome libraries were constructed. Quantification was performed using a Qubit 4.0 (Thermo Fisher, Waltham, MA, USA). PE150 sequencing was performed on the Novaseq platform (Illumina, San Diego, CA, USA), and differential expression analysis was conducted using EdgeR. Immune-related gene sets from the InnateDB database were used as references [33], and heatmaps were generated using ggplot.

### 2.14. Flow Cytometry Analysis

Spleen lymphocytes from tumor-bearing mice were isolated under sterile conditions and prepared into single-cell suspensions. The spleen lymphocytes were stimulated with the neoantigen for 4 h at 37 °C, 5% CO_2_. Afterward, Brefeldin A (S-1536, Beyotime, Shanghai, China) and Phorbol Myristate Acetate (PMA, P6741, Solarbio, Beijing, China) were added to block protein transport, and the cells were further incubated for another 4 h. To block Fc receptors, cells were incubated with anti-mouse CD16/32 antibody (93, Biolegend, UK) for 15 min at 4 °C. The cells were then stained with fluorescently conjugated antibodies against CD4 (FITC, GK1.5, Biolegend, London, UK), CD8a (APC, S18018E, Biolegend, London, UK), and CD3 (APC/Cyanine7, 17A2, Biolegend, London, UK) for 30 min. Afterward, the cells were incubated in Cyto-Fast™ Fix/Perm Buffer (Biolegend, London, UK) at room temperature for 20 min and washed twice with Cyto-Fast™ Perm Wash Solution. Then, anti-mouse IFN-γ antibody (PE, XMG1.2, Biolegend, London, UK) was used to stain the cells for 20 min in the dark at room temperature. Cells were washed twice with PBS and resuspended in 500 μL PBS for flow cytometry analysis, and data were analyzed using the FlowJo 10.8.1.

### 2.15. In Vivo Safety Evaluation

Female C57BL/6 mice were randomly divided into groups, with four mice per group. Every 7 days, 100 μg of pCSP, pCP, pP, or physiological saline was administered intramuscularly as a control, with the treatment continuing for 3 weeks. Serum from mice was collected 1 week after the final immunization via tail vein blood collection, with the mice being anesthetized prior to blood collection. After collection, the samples were centrifuged at 4 °C, 3000× rpm for 15 min, and the supernatants were analyzed using an automatic biochemical analyzer (BK-280, BIOBASE, Jinan, China) to measure the expression of alanine aminotransferase (ALT), aspartate aminotransferase (AST), lactate dehydrogenase (LDH), and blood urea nitrogen (BUN). After blood collection, the mice were euthanized using the same method as described in the previous section. The major organs, including the heart, liver, spleen, lungs, and kidneys, were fixed in 4% paraformaldehyde, paraffin-embedded, and then stained with hematoxylin and eosin (H&E). Subsequently, organ sections were scanned and photographed using a slide scanner (VS200, Olympus, Tokyo, Japan).

### 2.16. Statistical Analysis

All statistical analyses were performed using GraphPad Prism 9.5. For analyses involving two groups, *t*-tests (and nonparametric tests) were employed. When analyzing three or more groups, one-way ANOVAs (and nonparametric or mixed models) were used. Unless otherwise indicated, results are presented as the mean ± standard deviation. Statistical significance is denoted as * *p* < 0.05, ** *p* < 0.01, *** *p* < 0.001, and **** *p* < 0.0001.

## 3. Results

### 3.1. Construction of the DNA Vaccine pCSP and Fusion Protein Validation

To screen tumor neoantigen sequences specific to Panc02 cells, we used pVACtools to analyze genomic data and predicted 13 candidate mutations, with the corresponding mutated peptides expected to bind C57BL/6 mouse-specific MHC class I (H-2Kb and H-2Db). We integrated these 13 mutation sequences into a multi-epitope antigen sequence and performed codon optimization to obtain the DNA sequences encoding the tumor neoantigens, designed for efficient translation in mice (Figure 2A). Based on the research hypothesis, we inserted the SARS-CoV-2 RBD-encoding sequence into the 43rd amino acid position (aa 238) of the mouse CD63 protein, ensuring that the RBD remained positioned on the extracellular side when the fusion protein is assembled into EVs. Simultaneously, the multi-epitope neoantigen sequence was linked to the C-terminal of CD63 through the HHHD linker. To validate the design, we also constructed four control fusion proteins for comparison; namely, CP, CS, P, and C (Figure 2B). To verify the accuracy of the constructed plasmid sequences, we then performed restriction enzyme digestion on the synthesized plasmids and verified the accuracy of the target sequence using DNA agarose gel electrophoresis (Figure 2C). The plasmids were subsequently sequenced to confirm the integrity of their nucleotide sequences. To further validate the 3D structure of the fusion protein, we performed three-dimensional modeling using AlphaFold2 (Figure 2D). The results indicated that the transmembrane region of CD63 consists of four β-helix structures, and the RBD is located on the extracellular side of the transmembrane region. The multi-epitope neoantigen sequence is positioned on the opposite side via the linker, which is consistent with our design expectations. Additionally, referring to the structural analysis of SARS-CoV-2 RBD by Gupta et al. [34], we found that the binding sites of the RBD for DC-SIGN/R and ACE2 were not obstructed by other structural domains, providing good accessibility for the binding sites. The RBD is located far from other structures and remains structurally unaffected. Therefore, the design of the CSP fusion protein effectively preserves the spatial openness of the RBD, allowing efficient binding with DC-SIGN/D-SIGNR or ACE2. To further confirm the transmembrane topology of the CSP protein, we predicted its structure using the DeepTMHMM tool (Figure 2E). The results showed that all four transmembrane regions of CD63 were intact, the RBD was correctly located in the extracellular region, and the multi-epitope neoantigen sequence was positioned in the intracellular region. These findings demonstrate that the CSP fusion protein has the structural basis to assemble into EVs and exert its targeted function.

### 3.2. Preparation and Characterization of CSP-Modified EVs

To validate that cells transfected with pCSP successfully express the fusion protein, we first measured the mRNA expression levels of key components of CSP—mouse CD63, the neoantigen (ANT), and RBD—after transfection with blank plasmid, pCS, pP, pCP, or pCSP in HEK293T cells. The results showed that mouse CD63 gene expression was detected only in the CS, CP, and CSP groups; expression of the neoantigen fragment ANT was detected only in the CP and CSP groups; RBD expression was detected only in the CS and CSP groups. Given that HEK293T cells are of human origin, we further assessed the mRNA expression levels after transfecting the plasmids into mouse Panc02 cells. The results indicated that mouse CD63 was expressed in all groups, with the expression levels in the CS, CP, and CSP groups being approximately twice those of the blank control and P groups. The expression patterns of ANT and RBD were consistent with those in HEK293T cells (Figure 3A). These results demonstrate that the CSP sequence is effectively transcribed in both mouse and human cells after transfection with the pCSP plasmid.

To facilitate large-scale production of EVs expressing CSP, we transfected the pCSP plasmid into HEK293T cells and cultured them in exosome-specific medium for 48 h. Subsequently, the cell culture supernatants were collected, and EVs were extracted for further experiments. The concentration of the isolated EVs is approximately 7 × 10^7^ particles per µg of protein. Transmission electron microscopy (TEM) observations revealed that EVs isolated from pCSP-transfected cells exhibited the typical bilayer membrane structure characteristic of EVs (Figure 3B). Further magnification confirmed their integrity and purity, proving that the isolated EVs were of good quality. Nanoparticle tracking analysis (NTA) of the EVs from different plasmid transfections showed that the average particle size for all groups was between 100 and 150 nm. This indicates that transfection with the CSP plasmid did not affect the normal size distribution of the EVs, suggesting that the expression of CSP did not significantly interfere with the basic function and structure of EVs, and the formation and physical properties of the vesicles were maintained (Figure 3C).

To confirm whether CSP was successfully incorporated into extracellular vesicles, we performed Western blot analysis of EVs isolated from transfected cells. The results showed significant expression of the SARS-CoV-2 RBD, CD63, CD81, and TSG101 proteins in the EVs, while the endoplasmic reticulum marker protein Calnexin was only detected in the cell lysates and not in the EVs, further confirming the high purity and specific composition of the extracted EVs. Moreover, Western blot results also showed specific expression of RBD and CD63 in the EVs, indicating that CSP was successfully packaged into the EVs (Figure 3D). Additionally, both in the cell lysates and the EVs, the expression of CD63 in the C group was significantly higher than in the other groups, suggesting that modification of the CD63 protein reduced its expression levels. In the CS, CP, and CSP groups, the molecular weight of the CD63 protein increased, and the bands were observed at different positions based on the size of the fusion protein. Furthermore, corresponding bands for SARS-CoV-2 RBD and CD63 at the expected molecular weight were observed on the PVDF membrane in the CS and CSP groups (as indicated by black arrows in Figure 3D). These results collectively demonstrate that the CSP fusion protein is successfully expressed and loaded onto EVs.

### 3.3. CSP Modification Enhances EV Targeting to DCs

To evaluate whether CSP modification enhanced the targeting and uptake of EVs by DCs, we labeled the extracted EVs with PKH67 dye and co-incubated them with DC2.4 cells to evaluate their uptake using flow cytometry. The results showed that EVs derived from pCS- and pCSP-transfected cells were significantly more efficiently taken up by DC2.4 cells than in the other treatment groups. This indicates that the insertion of SARS-CoV-2 RBD into CD63 significantly enhanced the uptake of EVs by DC2.4 cells (Figure 4A). To further assess whether EV uptake led to DC activation, we measured cytokine secretion as the functional readout. Compared with other treatment groups, the CSP group significantly enhanced the secretion of TNF-α (Figure 4B) and IL-6 (Figure 4C). These findings suggest that CSP-modified EVs can not only efficiently enter DC2.4 cells but also activate them to initiate immune responses.

### 3.4. The pCSP Vaccine Enhances In Vitro Cytotoxicity and Immune Cell Activation

To evaluate the specific CTL response induced by the pCSP vaccine, we used spleen cells from immunized mice as effectors and Panc02 cells as target cells (Figure 4D,E). Compared with the control group and CS group, spleen cells from mice in the P, CP, and CSP groups showed higher levels of CTL activity. Notably, spleen cells from the CSP group exhibited significantly higher CTL activity than other groups, with significant antitumor effects at various effector-to-target ratios (12.5:1, 25:1, and 50:1), suggesting that CSP-modified EVs significantly enhanced the CTL activity of mouse spleen cells. To further assess cytokine regulation during the antitumor response after vaccination, we measured cytokine expression in spleen cells post-immunization. The results indicated that the CSP group significantly enhanced the secretion of TNF-α, IFN-γ, and IL-2, while there was no significant difference in IL-6 levels (Figure 4F). These findings suggest that the pCSP vaccine preferentially promotes the development of Type 1 Helper T (Th1) cells.

### 3.5. In Vivo Antitumor Effect of pCSP

In the tumor-bearing mouse model, we observed that pP, pCP, and pCSP all suppressed tumor growth compared with the control group, indicating that the neoantigens themselves can elicit antitumor responses in mice. Among these groups, the pCSP vaccine showed the most pronounced inhibitory effect, as evidenced by both tumor volume (Figure 5C) and tumor weight (Figure 5B,D), demonstrating its superior ability to delay tumor progression. In addition to administering the vaccine via intramuscular injection, we found that subcutaneous injection also achieved similar therapeutic effects (Appendix A), further demonstrating the antitumor efficacy of pCSP. Furthermore, the proportion of CD8+ cells in the spleens of the mice in the CSP group was significantly increased (Figure 5F), while the proportion of CD4+ T cells showed no significant changes. The analysis of IFN-γ secretion by CD8+ T cells further confirmed that the pCSP vaccine significantly enhanced the activity and immune infiltration of effector T cells in the tumor microenvironment (Figure 5E).

Heatmap analysis of immune-related genes (Figure 5F) revealed significant changes in the expression of several immune regulatory genes in the CSP group, suggesting that the pCSP vaccine exerts its antitumor effect by modulating multiple immune signaling pathways. To gain a deeper understanding of the impact of CSP treatment on immune responses, we further evaluated the expression changes in immune-related genes in detail (Figure 5I). We found that after CSP treatment, the expression of several immune-related genes was significantly altered, with genes such as Cd8a, Ifng, Il12rb1, Mmrn1, and Muc4 being upregulated, while genes such as Alox15, Ccr10, and Gp1bb were downregulated. The expression patterns of these genes suggest that CSP may influence immune responses and inflammatory reactions by modulating immune signaling pathways. In conclusion, the pCSP vaccine effectively inhibits tumor growth in vivo by enhancing CD8+ T cell activity and regulating the expression of immune-related genes.

### 3.6. Safety Evaluation of the pCSP Vaccine

To assess the safety of the pCSP vaccine, we evaluated histological changes in the heart, liver, spleen, kidneys, and lungs of mice through H&E staining (Figure 6A). The results showed no significant abnormalities in the heart, liver, and kidneys, while in the lung tissue, the CSP group exhibited localized pathological changes, including significant inflammatory cell infiltration in the alveoli and interstitial regions, as well as noticeable thickening of the alveolar septa. These changes may be associated with immune activation or inflammatory responses triggered by the pCSP plasmid injection, indicating potential mild inflammatory side effects that require further study and attention. In spleen H&E staining, the white pulp of the CSP group was significantly enlarged (Figure 6B), indicating that the pCSP vaccine enhanced the proliferation and immune activation of spleen lymphocytes. The body weight change curve (Figure 6C) showed no significant differences between the groups, suggesting that the vaccine injection did not have a significant impact on the growth of the experimental animals. Serum biochemical analysis (Figure 6D) showed significantly elevated ALT and AST levels in the CSP group, suggesting potential liver damage; increased BUN levels indicated possible mild kidney dysfunction; no significant differences in LDH levels between groups indicated minimal cellular damage. In conclusion, the pCSP vaccine showed overall good safety at the tested dose, although it may induce mild pulmonary inflammation and slight liver and kidney dysfunction. Further studies are needed to evaluate the long-term safety and potential toxic mechanisms of this vaccine.

## 4. Discussion

MANAs have significant therapeutic potential in cancer immunotherapy, particularly by inducing cancer-specific CTL responses and neoantigen-specific immune responses. This mechanism has been confirmed in several studies [35,36]. These neoantigens are caused by mutations in tumor cells and are presented by MHC molecules, effectively activating the immune system. The pVACtools utilized in this study provides a computational modular workflow toolkit for neoantigen prediction and prioritization (pVACseq) and the design of DNA vector-based vaccines (pVACvector). The pVACseq included eight MHC Class I prediction algorithms—including NetMHCpan, NetMHC, NetMHCcons, PickPocket, SMM, SMMPMBEC, MHCflurry, and MHCnuggets—and it predicts neoantigen peptide lengths of 8-11. The results from pVACtools analyses were considered robust, and the designed cancer vaccines have already been utilized in many cancer immunology studies, as well as in several ongoing clinical trials [29].

To achieve optimal CTL activity, these antigens need to be selectively presented to APCs—particularly DCs—which is crucial for enhancing antitumor immune responses [37,38,39]. To improve the delivery efficiency of these antigens, several techniques have emerged in recent years. For example, nanoparticle delivery systems [40] can enhance the targeted delivery of DNA vaccines to specific cells. Additionally, fusing targeted ligands or single-chain antibodies with the DNA sequence encoding antigens can enable direct delivery of the expressed neoantigen protein to target cells, further optimizing the immune response [39,41]. Such methods have shown significant promise in improving the immune efficacy of vaccines. In this study, we hypothesize that the neoantigens expressed by DNA vaccines can be targeted and delivered via EVs secreted by the host cell. We constructed a CD63-mediated multi-epitope neoantigen DNA vaccine and evaluated its immune activity and safety both in vitro and in vivo by fusing the SARS-CoV-2 RBD fragment with the neoantigen. Our study shows that the pCSP vaccine successfully delivers the encoded neoantigens via EVs to DCs and enhances their immune function. Specifically, pCSP vaccination significantly inhibited the growth of Panc02-derived tumors, demonstrating its strong antitumor immune potential.

The addition of the SARS-CoV-2 RBD fragment was intended to enhance the targeted delivery of the neoantigen through its high-affinity binding to DC-SIGN. DC-SIGN is an important C-type lectin receptor on dendritic cells that mediates antigen internalization and facilitates antigen presentation [42,43]. Through this targeted delivery, the pCSP vaccine improved the antigen uptake efficiency, optimized MHC-I antigen presentation, and ultimately enhanced CTL-mediated antitumor immune responses. We referred to the work of Gu et al. [44] on the adaptation of SARS-CoV-2 for use in mouse models and extracted the RBD fragment. Through the predicted 3D topology analysis of CSP, we believe that the fusion protein maintains its intended targeting function. It should be noted that the binding of CSP-modified EVs to specific targets has not yet been experimentally validated. In future work, we will investigate the interaction between EV-associated CSP and dendritic cell surface receptors to elucidate the underlying mechanistic relationships more precisely.

We extracted EVs from pCSP-transfected HEK293T cells. HEK293T cells are commonly used mammalian cell lines due to their high transfection efficiency and robust protein expression capabilities, making them widely applied in EV research. Previous studies have utilized HEK293T cells for the production of EVs for the immunization of mice [45,46,47]. Moreover, experiments have confirmed that there is no significant difference between EVs produced by human cell lines and those produced by mouse cell lines, and this does not affect antigen-specific responses [16]. Therefore, we considered using EVs produced by HEK293T cells to load the fusion protein CSP, which we confirmed with Western blot analysis (Figure 3D). The EVs isolated from these cells were further characterized using TEM and NTA. The results showed that CSP modification did not significantly alter the size, morphology, or particle size distribution of EVs, indicating that the modification did not interfere with the basic structure and function of EVs. Western blot analysis further confirmed that CSP was successfully incorporated into EVs. Additionally, in the Western blot analysis, we further observed the impact of CSP modification on the molecular weight of the CD63 protein. The results showed that the molecular weight of CD63 was increased in the CS, CP, and CSP groups compared with the C group. However, the CP group exhibited a significantly higher molecular weight band, which also showed smearing similar to that of the C group. We hypothesize that this smearing could be due to protein translation modifications, such as glycosylation, affecting the detection. In contrast, the smearing was less pronounced in the CS and CSP groups, suggesting that the insertion of the RBD fragment may have impacted the glycosylation modification of CD63. The precise consequences of this modification are not yet fully understood. Some studies suggest that the introduction of targeting sequences into the exosome membrane may affect the normal function of exosomal membrane proteins [48]. Therefore, our future research will focus on assessing the specific changes in glycosylation modifications, aiming to explore whether CSP modification influences the function of CD63, especially in the context of exosome-mediated antigen delivery.

DC-SIGN expression was significantly higher in DC2.4 cells, while SIGNR1 expression was higher in RAW264.7 cells (Appendix A). SIGNR1 in mice serves as the homolog of human L-SIGN (also known as CD209L), sharing structural and functional similarities [49]. Consistent with these findings and prior reports of RBD–DC-SIGN/L-SIGN interactions, we hypothesize that the incorporation of the RBD fragment enhances EV engagement with dendritic-cell lectin receptors (e.g., DC-SIGN), thereby facilitating receptor-mediated internalization. However, we acknowledge that the current evidence for RBD-facilitated endocytosis is indirect; in future work, we will validate the mechanism by knocking down DC-SIGN and by testing vaccine constructs bearing mutant RBD sequences to determine whether CSP-modified EV internalization is accordingly reduced. In vitro experiments confirmed that the pCSP vaccine significantly enhanced the targeting and uptake of EVs by DC2.4 cells while also activating the secretion of TNF-α and IL-6, indicating its potential to enhance antigen presentation. At present, we have only examined TNF-α and IL-6 expression, and future studies will further analyze additional cytokines such as IL-12 to better characterize DC activation following EV uptake.

Further in vivo studies demonstrated that the pCSP vaccine significantly inhibited tumor growth in Panc02 tumor-bearing mice, increased the proportion of CD8+ T cells in the spleen, and boosted the secretion of IFN-γ, while also enhancing the immune activity of effector T cells in the tumor microenvironment. We observed a preliminary therapeutic effect in vivo, as pP, pCP, and pCSP all suppressed tumor growth, indicating that neoantigens themselves can trigger antitumor responses in mice. Furthermore, linking CD63 with neoantigens allowed antigen delivery through EVs, thereby enhancing the antitumor effect [16], and the addition of the RBD sequence to the extracellular domain of CD63 further strengthened the vaccine’s inhibitory activity against tumors. Future studies will also assess additional CD8+ T-cell cytotoxicity markers such as granzyme B and CD107a to reinforce these conclusions. Previous studies have shown that DNA vaccines are efficiently absorbed and expressed by muscle cells [50] and DCs [51]. DCs are the most effective antigen-presenting cells, playing a critical role in triggering both CD4+ and CD8+ T cell responses through antigen processing and presentation [52]. By targeting DCs through EVs, neoantigens can be delivered, allowing muscle cells transfected with the DNA vaccine to indirectly present antigens. This design not only simplifies the administration of DNA vaccines, but also enhances their immunogenicity.

It is known that angiotensin-converting enzyme 2 (ACE2), neuropilin-1 (NRP-1), and DC-SIGN-related receptor (DC-SIGNR, also known as L-SIGN) interact with the SARS-CoV-2 spike protein and serve as the functional receptors for SARS-CoV-2 entry [53,54]. Although those receptors are also expressed on dendritic cells, they could drive pathogen antigen recognition, dendritic cell function, and neuroantigen-specific immune responses. ACE2, NRP1, and L-SIGN are also expressed by various cells in the body [20,55,56]. After local delivery of the pCSP vaccine, some self-assembled EVs would be disseminated into the bloodstream; they carry an RBD region that binds to receptors, which may lead to off-tumor effects resulting from cell surface presentation of MHC-I and a peptide antigen complex that is recognized and attacked by activated T cells. We evaluated the safety of the pCSP vaccine and found mild liver and kidney function damage or noticeable pulmonary inflammation in vaccinated mice. Meanwhile, as the RBD itself is a key immunogenic component of SARS-CoV-2, pre-existing anti-SARS-CoV-2 immunological memory potentially decreases engineered EV expression and limits antitumor immune effects [57]. The RBD may also engage host receptors that are not restricted to DCs, creating plausible avenues for off-target binding and unanticipated biodistribution. To address those points, we are currently exploring alternative fragments to target to improve specificity, such as by including virus peptides from pathogens that are relatively less prevalent and that are only reported to interact with DC-SIGN but not with L-SIGN for cell entry. The candidate peptides will be further screened with an AI-assisted program, “alphafold-peptide-receptors”, that identifies and ranks peptide–receptor interactions. To reduce dual-use and off-target liabilities while preserving DC uptake, RBD-free DC-targeting modules can also be substituted—including anti-DEC-205 (CD205) scFv or C-type lectin-directed ligands (e.g., mannose/fucose-based glycoconjugates)—which focus on delivery to professional antigen-presenting subsets. Together, these mitigation strategies provide a practical path to de-risking DC-targeted platforms that originally relied on RBD.

Overall, this proof-of-concept study proposes an innovative DNA vaccine design strategy that achieves the targeting of neoantigens to DCs via EVs through genetic modification of the extracellular domain of CD63. The assembly of the individualized neoantigens and EVs mediated various DC-targeted molecules occurs within the host cell, requiring only modification of the vaccine sequence to enable targeted delivery. Our findings provide new insights to enhance the targeted delivery capability of DNA vaccines and open new avenues for future vaccine design and immunotherapy approaches. This strategy demonstrates significant potential for the immunotherapy of tumors with low immunogenicity, such as pancreatic cancer, and it may facilitate the clinical translation of novel vaccine development technologies.

## 5. Conclusions

This study developed a novel DNA vaccine, pCSP, which utilizes the CD63 protein expressed on extracellular vesicles and enhances dendritic cell uptake through the SARS-CoV-2 RBD fragment. This process improves antigen presentation and CD8+ T cell activation. In vitro and in vivo experiments demonstrated the promising antitumor effects of the pCSP vaccine. However, mild liver and kidney dysfunction, as well as localized pulmonary inflammation, were observed. Future efforts should focus on further optimizing the dendritic-cell-targeting domain. In conclusion, this study provides a new approach for the targeted delivery of DNA vaccines in vivo.

## Figures and Tables

**Figure 1 vaccines-13-00977-f001:**
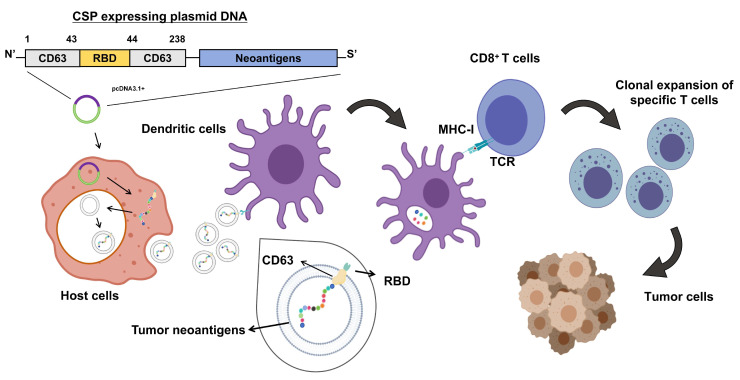
Schematic diagram of the antitumor immune mechanism of the DNA vaccine pCSP. After plasmid DNA is transfected into the host cells, the plasmid DNA is expressed within the host cell, producing a fusion protein that contains CD63 and the tumor neoantigen. These proteins are secreted via EVs (such as exosomes) and are delivered to DCs. After the DCs uptake the fusion protein, the tumor neoantigens are processed and presented to CD8+ T cells via MHC-I molecules. The T cell receptors (TCRs) specific to the neoantigens are activated, leading to the clonal expansion of specific CD8+ T cells. The activated T cells migrate to the tumor site, where they recognize and kill tumor cells expressing the corresponding antigens, thereby achieving antitumor immune effects. Created in MedPeer. G.L. (2024) https://www.medpeer.cn.

**Figure 2 vaccines-13-00977-f002:**
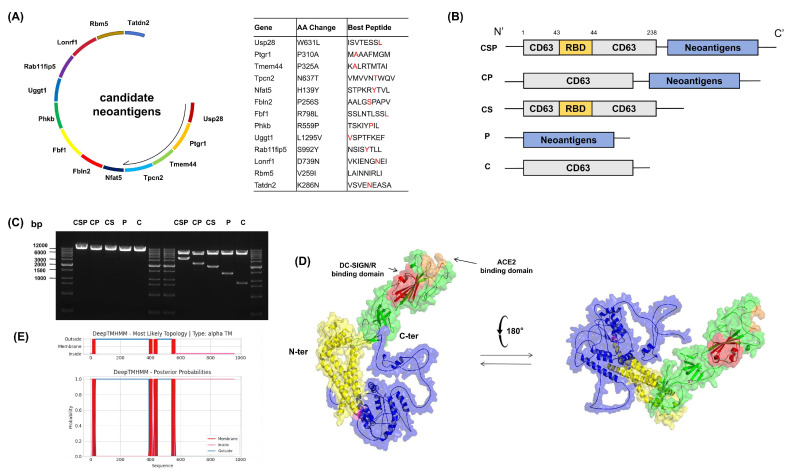
Construction of the CD63 fusion multi-epitope neoantigen DNA vaccine pCSP combined with SARS-CoV-2 RBD. (**A**) Selection of tumor-specific neoantigen epitopes. Based on bioinformatic prediction and optimization, 13 candidate Panc02-derived neoantigen genes were identified. The table lists gene names, amino acid substitutions, and optimized peptide sequences. (**B**) Schematic diagram of the fusion construct CSP and its control proteins (CP, CS, P, and C). CSP incorporates CD63, Panc02 neoantigens, and the SARS-CoV-2 RBD. (**C**) Verification of recombinant plasmids by agarose gel electrophoresis. Left: BamHI single digestion; right: XbaI and BamHI double digestion, confirming correct insertion and orientation of constructs. (**D**) Structural modeling of the CSP fusion protein. Three-dimensional models were generated using PyMOL. Blue: neoantigen domain; yellow: CD63 domain; green: RBD; purple: linker; red: DC-SIGN/R binding domain; orange: ACE2 binding domain. Multiple views illustrate the overall conformation of the protein. (**E**) Transmembrane topology prediction of CSP using the DeepTMHMM algorithm. The diagram indicates intracellular, transmembrane, and extracellular regions, consistent with the expected localization for EV surface display.

**Figure 3 vaccines-13-00977-f003:**
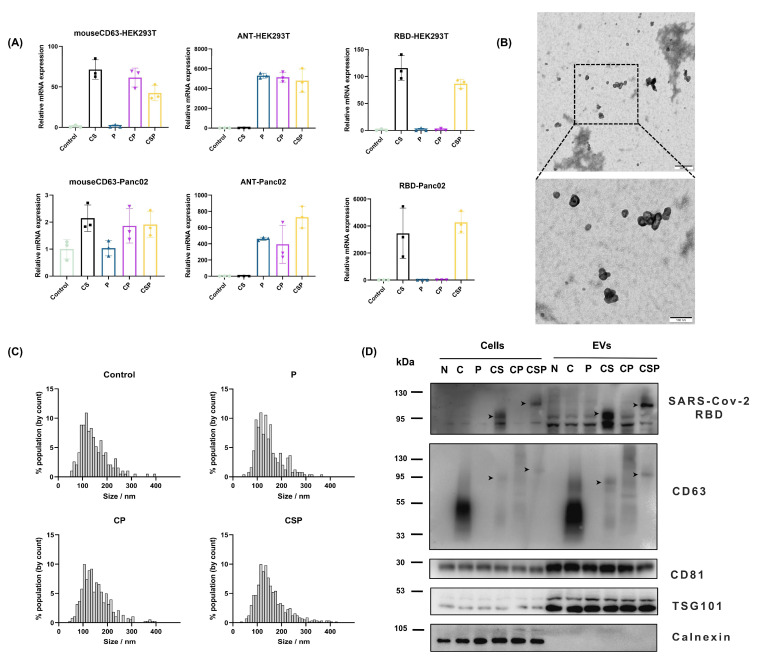
Characterization of EVs expressing the CSP fusion protein. (**A**) Gene expression profiling of plasmid-transfected cells. HEK293T and Panc02 cells were transfected with pCSP or derivative plasmids (pC, pCS, pP, pCP), and mRNA expression levels of mouse CD63, tumor neoantigen (ANT), and RBD were quantified using qRT-PCR (*n* = 3, Mean ± SD). (**B**) Morphological analysis of EVs secreted by pCSP-transfected cells using transmission electron microscopy (TEM). EVs displayed a typical cup-shaped morphology. Scale bar = 100 nm. (**C**) Nanoparticle tracking analysis (NTA) showing the size distribution profile of EVs, confirming a mean diameter consistent with typical exosomes. (**D**) Western blot analysis of cellular and EV lysates for protein markers. Expression of RBD and CD63 confirmed successful incorporation of the fusion protein into EVs, while CD81 and TSG101 served as EV markers; Calnexin was used as a negative control to exclude cellular contamination. The black arrow indicates the position of the fusion protein detected with the CD63 antibody, which can also be recognized by the RBD antibody at the same molecular weight.

**Figure 4 vaccines-13-00977-f004:**
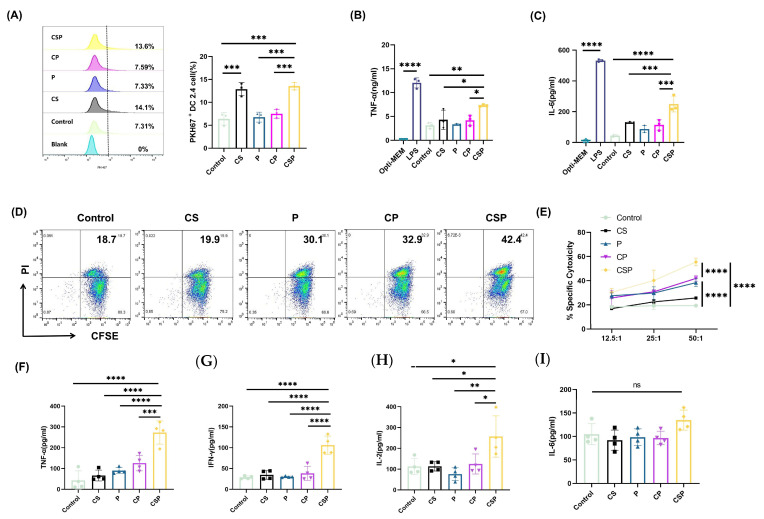
pCSP Plasmid enhances the activity of antigen-presenting cells and tumor cell killing. (**A**) Flow cytometric analysis of antigen uptake by DC2.4 cells. The percentage of PKH67-positive cells indicates EV internalization efficiency, with representative plots (left) and quantitative analysis (right). (**B**,**C**) Cytokine secretion by DC2.4 cells after 8 h of co-incubation with EVs derived from DNA-vaccine-transfected cells; the secretion levels of cytokines TNF-α and IL-6 in the supernatants of DC2.4 cells (*n* = 3). Levels of (**B**) TNF-α and (**C**) IL-6 in cell culture supernatants were measured using ELISA. (**D**,**E**) Cytotoxic T lymphocyte (CTL) activity against Panc02 tumor target cells. (**D**) Representative flow cytometry plots showing CFSE-labeled target cells and PI-positive dead cells at an effector-to-target ratio of 25:1. (**E**) Quantitative analysis of specific cytotoxicity at different effector-to-target ratios (12.5:1, 25:1, 50:1; *n* = 3). (**F**–**I**) Cytokine production by splenic lymphocytes following co-culture with tumor cells; the levels of cytokines (**F**) TNF-α, (**G**) IFN-γ, (**H**) IL-2, and (**I**) IL-6 were measured (*n* = 4). * *p* < 0.05, ** *p* < 0.01, *** *p* < 0.001, and **** *p* < 0.0001.

**Figure 5 vaccines-13-00977-f005:**
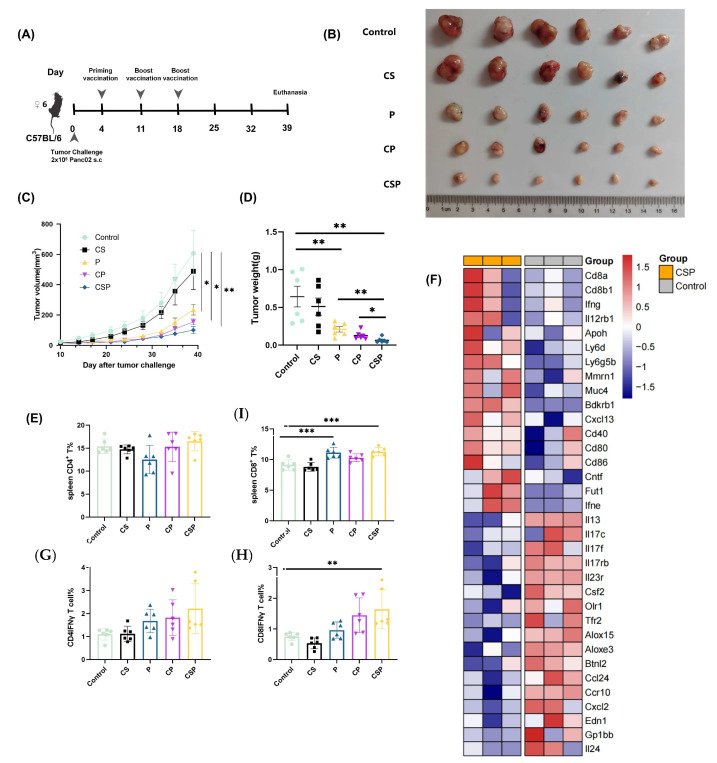
pCSP inhibits tumor growth and modulates immune responses in C57BL/6 mice. (**A**) Vaccination and tumor challenge protocol for C57BL/6 mice. Initial vaccination and two booster immunizations were performed on days 4, 11, and 18 after tumor inoculation, with mice euthanized on day 39 for sample collection (*n* = 6). (**B**) Representative images of excised tumors from each treatment group. (**C**) Tumor growth curves, showing that tumor volumes were significantly reduced in the P, CP, and CSP groups compared with controls, with the CSP group exhibiting the greatest inhibition of tumor progression. (**D**) Tumor weight at the endpoint confirmed reduced tumor burden in vaccinated groups, with the CSP group showing the most significant reduction. (**E**–**H**) Flow cytometric analysis of splenic immune cell populations: (**E**) proportion of CD4+ T cells; (**F**) proportion of CD8+ T cells; (**G**) proportion of CD4+ IFN-γ+ T cells; (**H**) proportion of CD8+ IFN-γ+ T cells. (**I**) Heatmap displaying differential expression of immune-related pathway genes in spleens of CSP-treated mice compared with controls. * *p* < 0.05, ** *p* < 0.01, and *** *p* < 0.001.

**Figure 6 vaccines-13-00977-f006:**
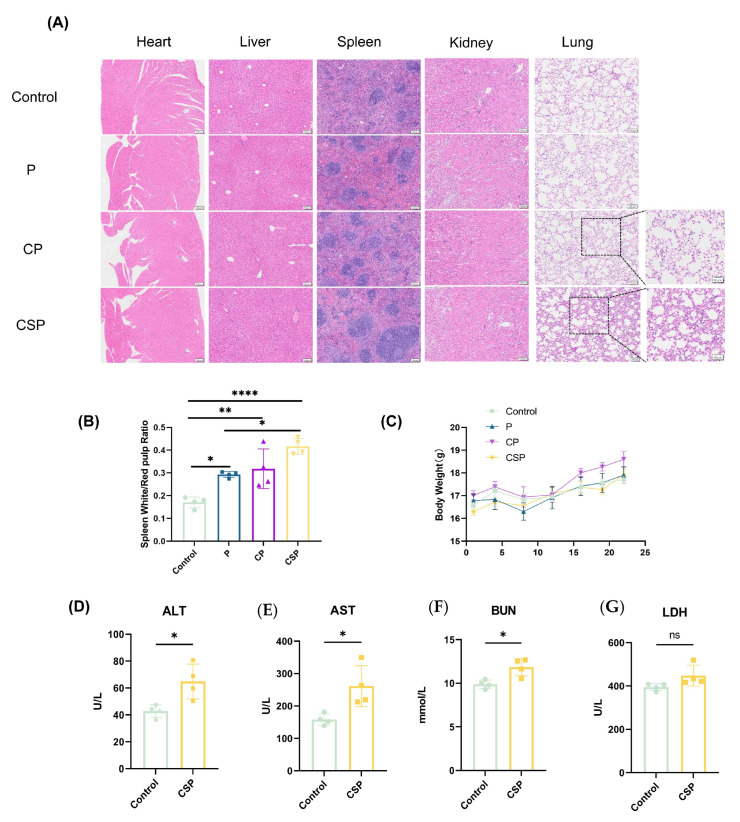
Effects of pCSP plasmid injection on organ pathology and physiological parameters. (**A**) Representative H&E staining of major organ sections (heart, liver, spleen, kidneys, and lungs) collected from mice in each treatment group. Scale bar = 100 μm. The area of inflammatory infiltration in the lungs is highlighted with a black box. (**B**) Quantitative analysis of the ratio of white pulp to red pulp area in the spleen. Mice in the CSP group exhibited significant white pulp proliferation compared with the controls. (**C**) Body weight change curve for mice in each group. (**D**–**G**) Serum biochemical indices reflecting hepatic, renal, and tissue injury were measured: (**D**) alanine aminotransferase (ALT, U/L), (**E**) aspartate aminotransferase (AST, U/L), (**F**) blood urea nitrogen (BUN, mmol/L), and (**G**) lactate dehydrogenase (LDH, U/L). Data are presented as the mean ± SEM. * *p* < 0.05 compared with the control group, ns *p* > 0.05, ** *p* < 0.01, and **** *p* < 0.0001.

## Data Availability

The RNA-Seq data generated in this study have been deposited in the NCBI Sequence Read Archive (SRA) under accession number PRJNA1199780. All other data analyzed in this study are available from the corresponding author upon reasonable request. Requests to access these datasets should be directed to Jibin Dong (jbdong@fudan.edu.cn).

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
