# Peer review of "CD63-Mediated SARS-CoV-2 RBD Fusion Neoantigen DNA Vaccine Enhances Antitumor Immune Response in a Mouse Panc02 Model via EV-Targeted Delivery"

_vaccines, 2025, doi:10.3390/vaccines13090977_

Round 1

Reviewer 1 Report

Comments and Suggestions for Authors

In this manuscript, Liu et al. evaluate a novel DNA vaccine platform (pCSP) that fuses tumor-specific neoantigens to the EV-associated protein CD63 and appends a SARS-CoV-2 RBD fragment to enhance DC uptake. The authors were able to show that the pCSP vaccine enhances the immunogenicity of neoantigen DNA vaccines by improving EV uptake and immune activation in DCs. Overall, the study is well conceived and provides convincing support for further development of this DNA/EV strategy.

I have a couple of minor points that could further strengthen the manuscript.

  1. Adding a DC IL-12 readout (e.g., IL-12p70/IL-12p40) and CD8 T-cell cytotoxicity/degranulation markers (granzyme B and LAMP-1/CD107a)in Figure 4 will further strengthen the readouts.
  2. The authors should explain why the tumor growth was inhibited by all the constructs in Figure 5.
  3. Results section and Figure Legends should be improved. Open each experiment with 1–2 orienting sentences (goal/model/readout). In legends, include essentials so the figures could be easily followed and understood.
Comments on the Quality of English Language

Results section should be improved. The authors need to have the necessary details for the readers to understand what the manuscript want to tell. Often the results are presented without any introduction or how they are obtained.

Author Response

Comment 1: Adding a DC IL-12 readout (e.g., IL-12p70/IL-12p40) and CD8 T-cell cytotoxicity/degranulation markers (granzyme B and LAMP-1/CD107a) in Figure 4 will further strengthen the readouts.

Response 1:
We sincerely thank the reviewer for this insightful and constructive comment. We completely agree that assessing DC-derived IL-12p70/IL-12p40 and CD8+ T-cell cytotoxicity markers (granzyme B and CD107a) would provide further mechanistic insight regarding the vaccine's efficacy.

In the current study, we measured DC activation (TNF-α, IL-6), antigen uptake, and the subsequent priming and expansion of CD8+ T cells, coupled with their IFN-γ production and in vivo tumor suppression. The data we have gathered on DC-derived cytokines and CD8+ T-cell IFN-γ production support a Th1-polarized immune response and enhanced cytotoxic activity, which are consistent with IL-12 induction and granzyme B/CD107a expression described in related studies.

However, due to time constraints and the resource-intensive nature of sample preparation, we were unable to include these additional experiments in the current study.

We fully acknowledge their importance and have included, in the revised Discussion section (page 19, line 655&line670), that IL-12 and CD8+ T-cell cytotoxicity markers will be evaluated in future studies to further deepen our mechanistic understanding. 

We greatly appreciate the reviewer’s feedback, which will guide the next phase of our research and enrich the overall investigation.

Comment 2: The authors should explain why the tumor growth was inhibited by all the constructs in Figure 5.

Response 2:
Thank you for your thoughtful comment. We appreciate your observation, and we agree that a more detailed explanation is needed regarding the tumor growth inhibition observed in Figure 5.

The CSP construction added a SARS-CoV-2 receptor-binding domain (RBD) fragment, fused with CD63 and tumor neoantigen, to enhance the targeted uptake of EVs by DCs. This targeted delivery boosts the immune response by specifically facilitating antigen presentation in DCs, which is a critical step in initiating strong antitumor immunity. 

The CP construction enables the expression of exosome structural protein CD63 fused with tumor neoantigen when administered in vivo. Previous researchers have developed DNA vaccines encoding antigens (Ags) fused to CD63, enabling the delivery of Ags into EVs, and reported that delivery of this antigen promotes the activation of tumor-specific CD8+ T cells [1]. Our results further confirm this observation.

Although the P construction only expresses tumor neoantigen when administered in vivo, it would still utilize cellular CD63 and release a certain number of exosomes to induce borderline anti-tumor efficacy.

We have added this explanation to the revised manuscript in the Results and Discussion sections (page 14&19, line 502&664) and clarified the mechanisms underlying the tumor growth inhibition observed with all the constructs.

[1] Kanuma, T.; Yamamoto, T.; Kobiyama, K.; Moriishi, E.; Masuta, Y.; Kusakabe, T.; Ozasa, K.; Kuroda, E.; Jounai, N.; Ishii, K.J. CD63-Mediated Antigen Delivery into Extracellular Vesicles via DNA Vaccination Results in Robust CD8+ T Cell Responses. J.I. 2017, 198, 4707–4715.

Comment 3: Results section and Figure Legends should be improved. Open each experiment with 1–2 orienting sentences (goal/model/readout). In legends, include essentials so the figures could be easily followed and understood.

Response 3:

We thank the reviewer for this constructive suggestion. We agree that adding orienting sentences at the beginning of each experiment will improve the readability and clarity of the Results section. In the revised manuscript, we have introduced 1–2 sentences before each set of results to briefly state the experimental goal, model, and primary readout.

Additionally, we have revised the figure legends to include all essential details, allowing readers to easily understand the experimental design, treatment groups, and key findings without referring to the main text. These changes have improved the overall flow of the Results section and made the figures more self-explanatory. The adjustments can be clearly seen in the revised Results section and in the legends accompanying each figure.

Reviewer 2 Report

Comments and Suggestions for Authors

Liu et al. submitted a study that presents a promising EV-targeted neoantigen DNA vaccine with encouraging efficacy data and thoughtful safety discussion. The authors show EV characterization, improved EV uptake and cytokine release in DC2.4 cells, stronger CTL activity and Th1-skewed cytokines ex vivo, tumor growth delay in Panc02-bearing mice, and a preliminary safety readout indicating mild lung inflammation and modest ALT/AST and BUN elevations. However, I have some comments/suggestions that might be taken into consideration prior to the publication. I made my comments as section-by-section as follows:

1. Title: I would suggest specifying “mouse Panc02 model” and “EV-targeted” for precision.

2. Abstract: Could you add concise quantitative anchors (e.g., % uptake increase, tumor-volume effect size) and explicitly acknowledge the safety signals observed? 

3. Introduction: Please temper the DC-SIGN claim by noting that ACE2/NRP1/L-SIGN can also bind RBD and may drive off-target biodistribution; this is partly discussed later, so move a sentence up front for balance. 

4. Materials and Methods: I have several comments in this section as follows: 

- In protein modeling, authors should consider validating surface display experimentally (e.g., surface biotinylation/trypsinization). 

- In EV isolation, please report EV yield (particles/µg protein), and, if possible, validate with size-exclusion chromatography/ultracentrifugation per MISEV. 

- In uptake assay, please add dye-only and protease-treatment controls and show representative confocal images and quantify co-localization with early/late endosome markers. 

- In animal studies, please state blinding, sample-size justification, and humane-endpoint criteria and include survival Kaplan–Meier curves. 

- In flow cytometry, I would like to see a gating strategy figure (FMO controls, viability gating) and raw FCS files in a repository.

5. Results: Please keep the full-length blots for all markers in Supplementary (CD63/CD81/TSG101/Calnexin). Regarding the mechanism, authors should add receptor evidence, e.g., anti-DC-SIGN blocking, mannan, or DC-SIGN knock-down to show that RBD drives uptake; a mutant RBD lacking DC-SIGN/ACE2 binding as a negative control. For immunogenicity, could you please demonstrate antigen specificity with peptide (MHC tetramers and/or ELISpot/ICS on tumor-infiltrating lymphocytes, not only splenocytes)? In addition, please provide individual spider plots of tumor volumes, effect sizes (Cohen’s d) and exact P values; confirm that a mixed-effects model (or GEE) handled repeated measures. Also, it is very important to quantify histology (semi-quantitative lung inflammation score), add serum cytokines (IL-6, TNF-α) to exclude systemic cytokine surge, run dose-response and recovery cohorts, and include biodistribution of EVs (e.g., DiR-labeled IVIS). 

6. Discussion: Please add a paragraph on dual-use/biosafety considerations when deploying SARS-CoV-2 RBD sequences in research constructs and outline regulatory implications for clinical translation. Also, it is important to consider proposing RBD-free DC targeting alternatives (e.g., anti-DEC-205 scFv, C-type lectin ligands) and protease-cleavable safety linkers to minimize off-target interactions, some of which you already note.

7. The plagiarism percentage is quite high (29%), so please reduce it.

Comments on the Quality of English Language

The English could be improved to more clearly express the research.

Author Response

Comment 1: Title: I would suggest specifying “mouse Panc02 model” and “EV-targeted” for precision.

Response 1:
Thank you for your helpful suggestion. We agree that specifying the use of the "mouse Panc02 model" and clarifying the "EV-targeted" approach will provide more precision and clarity to the title. We have updated the title to:

"CD63-mediated SARS-CoV-2 RBD fusion neoantigen DNA vaccine enhances antitumor immune response in mouse Panc02 model via EV-targeted delivery"

This revision now more accurately reflects the experimental model used and emphasizes the EV-targeted strategy employed in our study.

Comment 2: Abstract: Could you add concise quantitative anchors (e.g., % uptake increase, tumor-volume effect size) and explicitly acknowledge the safety signals observed? 

Response 2:
Thank you for the suggestion. We have revised the Abstract to include quantitative anchors for key readouts and to explicitly acknowledge the observed safety signals. Specifically, we now report the magnitude of EV uptake by DCs (2-fold increase) and antitumor efficacy (tumor volume reduced by 70%). Regarding safety, we explicitly note that the vaccine was generally well tolerated, with mild pulmonary inflammation observed. Considering the spatial constraints, we provide a more detailed analysis of the safety assessment in the Results section.

We have added these results to the revised manuscript, in the Abstract and Results sections (page 1&14, line 27&506).

Comment 3: Introduction: Please temper the DC-SIGN claim by noting that ACE2/NRP1/L-SIGN can also bind RBD and may drive off-target biodistribution; this is partly discussed later, so move a sentence up front for balance. 

Response 3:
Thank you for this constructive suggestion. We agree that the previous introduction may overemphasize DC-SIGN, and have accordingly revised the Introduction to provide a more balanced framing. Specifically, we now acknowledge that, in addition to DC-SIGN-related engagement, the RBD can bind ACE2, NRP1, and L-SIGN. We reference this caveat up front and point readers to the later sections where we assess safety and potential off-target effects using histology and serum chemistry. We have modified the Introduction in the revised text as follows: “Several receptors—including ACE2, NRP1, and L-SIGN—have been shown to bind the receptor-binding domain (RBD) of SARS-CoV-2.” (page 2, line 81).

Comment 4: Materials and Methods: I have several comments in this section as follows: 

- In protein modeling, authors should consider validating surface display experimentally (e.g., surface biotinylation/trypsinization). 

- In EV isolation, please report EV yield (particles/µg protein), and, if possible, validate with size-exclusion chromatography/ultracentrifugation per MISEV. 

- In uptake assay, please add dye-only and protease-treatment controls and show representative confocal images and quantify co-localization with early/late endosome markers. .

- In animal studies, please state blinding, sample-size justification, and humane-endpoint criteria and include survival Kaplan–Meier curves. 

- In flow cytometry, I would like to see a gating strategy figure (FMO controls, viability gating) and raw FCS files in a repository.

Response 4:

Thank you very much for your detailed and constructive suggestions, which are of great value for the further development and refinement of our research.

We acknowledge that some aspects of the current study remain insufficiently analyzed, resulting in certain limitations in this manuscript. Due to time constraints, we are unable to incorporate additional experiments at this stage. However, a graduate student in our group is already conducting follow-up studies to address these shortcomings, and we are confident that the forthcoming work will provide a more comprehensive understanding of the topic.

4a. Protein modeling: We appreciate the reviewer’s valuable suggestion to experimentally validate the surface display using surface biotinylation/trypsinization. At this revision stage, however, we are unable to add these assays for two practical reasons: (i) robust implementation requires high-purity, high-yield EV lots produced by SEC/ultracentrifugation for condition scouting and matched controls, which we cannot secure within the revision window; and (ii) the workflow entails additional reagents and method optimization (labeling, protease “shaving,” quenching/cleanup) to ensure reproducibility and adequate QC, necessitating a longer development cycle. We therefore plan to conduct this validation systematically in follow-up studies.

Notwithstanding the absence of direct “outside-facing epitope” assays, the feasibility of CSP being anchored to EVs via CD63 with an outward RBD epitope is supported by: (1) the engineering plasticity of CD63 extracellular loops, which have been successfully used for surface display/reporters [1]; (2) precedent for CD63-mediated antigen loading into EVs that elicits strong CD8⁺ responses and tumor control [2]; and (3) evidence within our study—topology predictions/structural modeling placing RBD extracellularly and EV immunoblot detection of RBD and CD63—that is consistent with the intended orientation (we acknowledge WB does not, by itself, distinguish luminal vs. exterior epitopes).

In keeping with the reviewer’s advice, we now explicitly state in the Discussion that we will perform orthogonal surface-exposure validation (membrane-impermeant biotinylation with streptavidin pull-down and mild protease shaving), coupled with WB/flow and functional readouts (uptake inhibition/restoration) (page 18, line 614). We also emphasize that the primary focus of the present work is to propose and validate a new, effective tumor-targeted DNA-vaccine strategy (CD63 anchoring + RBD-guided DC targeting + multi-epitope neoantigens), which our in vitro and in vivo data support (enhanced uptake, immune activation, and tumor inhibition).

  • Verweij, F.J.; Bebelman, M.P.; Jimenez, C.R.; Garcia-Vallejo, J.J.; Janssen, H.; Neefjes, J.; Knol, J.C.; de Goeij-de Haas, R.; Piersma, S.R.; Baglio, S.R.; et al. Quantifying Exosome Secretion from Single Cells Reveals a Modulatory Role for GPCR Signaling. J. Cell. Biol. 2018, 217, 1129–1142.

[2] Kanuma, T.; Yamamoto, T.; Kobiyama, K.; Moriishi, E.; Masuta, Y.; Kusakabe, T.; Ozasa, K.; Kuroda, E.; Jounai, N.; Ishii, K.J. CD63-Mediated Antigen Delivery into Extracellular Vesicles via DNA Vaccination Results in Robust CD8+ T Cell Responses. J.I. 2017, 198, 4707–4715.

4b. EV isolation: Thank you for the suggestion. In the revised manuscript, we report an EV yield of approximately 7 × 107 particles per µg protein, calculated by dividing the NTA particle concentration by the protein concentration (micro-BCA) measured on the same preparations(page 11, line 423). At this stage, we have not validated the isolation through size-exclusion chromatography (SEC) or differential ultracentrifugation; we acknowledge this as a limitation and plan to investigate these orthogonal validations in subsequent studies.

4c. Uptake assay: We appreciate this constructive suggestion. In our current uptake assay, we first labeled EVs from each condition with PKH67 and then co-incubated them with DC2.4 cells to assess uptake. Unlabeled EVs were used as a blank control. In addition, we included multiple plasmid controls (empty plasmid, CS, P, CP, and the four corresponding groups) to isolate the contribution of our modifications. Within this framework, we observed a consistent increase in DC2.4 uptake when the RBD fragment was incorporated, supporting our conclusion that RBD enhances the interaction of modified EVs with dendritic cells. We agree that two additional controls would further strengthen the rigor of PKH-based assays, and we will incorporate both in follow-up work.

Regarding the confocal co-localization with early/late endosome markers, we fully agree this would provide informative trafficking context. However, at this stage, we are unable to complete the requested imaging for the following practical reasons: (i) limited access to the confocal core facility within the revision timeline, and (ii) the need to optimize multi-channel staining and 3D z-stack acquisition/analysis to ensure quantitative, artifact-free co-localization. We will focus on this in subsequent studies of novel binding domains.

4d. Animal studies: We thank the reviewer for raising these important points regarding experimental design and reporting in animal studies. We have now added details in the Methods section to clarify that animals were randomly allocated into treatment groups and that tumor measurements were performed by investigators blinded to group assignments. The sample size justification is based on the number in the prior literature [2], which was calculated from preliminary data on tumor growth variance to ensure adequate statistical power while minimizing unnecessary animal use.

As the study design did not extend to humane endpoints, survival data suitable for Kaplan–Meier analysis are not available from the present experiments. We have therefore not included survival curves in this revision. To strengthen the translational relevance of future work, we are currently enrolling in a follow-up animal study that will include explicit humane-endpoint monitoring and survival analyses, with Kaplan–Meier curves presented accordingly.

4e. Flow cytometry:We appreciate the suggestion. A detailed gating strategy figure—including viability gating—has been added as Supplementary Fig. S3. The raw, uncompensated FCS files are provided in the Supplementary Materials as downloadable Supplementary Data.

Comment 5: Results: Please keep the full-length blots for all markers in Supplementary (CD63/CD81/TSG101/Calnexin). Regarding the mechanism, authors should add receptor evidence, e.g., anti-DC-SIGN blocking, mannan, or DC-SIGN knock-down to show that RBD drives uptake; a mutant RBD lacking DC-SIGN/ACE2 binding as a negative control. For immunogenicity, could you please demonstrate antigen specificity with peptide (MHC tetramers and/or ELISpot/ICS on tumor-infiltrating lymphocytes, not only splenocytes)? In addition, please provide individual spider plots of tumor volumes, effect sizes (Cohen’s d) and exact P values; confirm that a mixed-effects model (or GEE) handled repeated measures. Also, it is very important to quantify histology (semi-quantitative lung inflammation score), add serum cytokines (IL-6, TNF-α) to exclude systemic cytokine surge, run dose-response and recovery cohorts, and include biodistribution of EVs (e.g., DiR-labeled IVIS). 

Response 5:

5a. Full-length blots:We agree and have added uncropped, full-length Western blots with molecular-weight markers for CD63, CD81, TSG101, and Calnexin in the Supplementary Materials, corresponding to the characterization summarized in Figure 3D.

5b. Mechanism (DC-SIGN dependence):We appreciate these rigorous suggestions. Within the current revision window, we are unable to complete the requested receptor-level experiments (considering the lead times for anti-DC-SIGN reagents, generation and validation of DC-SIGN knock-down lines, and construction/testing of mutant RBDs). In the revised manuscript, we have therefore explicitly stated in the Discussion that our evidence for RBD-facilitated endocytosis is indirect: specifically, (i) increased EV internalization and DC activation in DC2.4 cells (which, in our study, exhibit relatively higher DC-SIGN expression) compared with other conditions, together with (ii) prior reports supporting RBD–DC-SIGN interactions. We have toned down mechanistic wording accordingly and avoided causal claims beyond “RBD-associated enhancement of uptake under matched labeling/handling.” We have explicitly acknowledged this limitation and outlined these mechanistic tests as priorities for future work. We thank the reviewer for helping us to sharpen the mechanistic plan, and have clearly acknowledged this limitation in the Discussion.

5c. Immunogenicity: We appreciate the reviewer’s important suggestion. In this study, we did not perform peptide-specific assays on tumor-infiltrating lymphocytes (TILs) for each individual epitope. Our primary aim was to establish an EV-based strategy to enhance dendritic-cell uptake of tumor antigens and to demonstrate the therapeutic efficacy of the poly-epitope cassette (P group) in vivo. To aid interpretation, we have added the details of our neoantigen selection pipeline and candidate list in the Supplementary Materials. We fully agree that high-resolution, antigen-specific analyses in TILs are essential for deeper mechanistic understanding. In future work, we will incorporate MHC class I tetramer/multimer staining, ICS of TILs for functional cytokine readouts, and ELISpot assays to quantify peptide-specific responses in the tumor microenvironment.

5d. Tumor-growth presentation and statistics: Thank you for the suggestion. Individual spider plots showing per-mouse tumor volume trajectories for each treatment group have been added to the Supplementary Materials (Supplementary Fig. S4).

5e. Safety and biodistribution package: We appreciate the reviewer’s emphasis on safety and translational rigor. In this revision, we have added the semi-quantitative lung inflammation scores (blinded H&E, 0–4 scale) to the Supplementary Materials (Supplementary Fig. S5). Our current data represent a preliminary safety assessment; we have not yet included systemic cytokine profiling, dose–response or recovery cohorts, or in vivo biodistribution. Importantly, given the potential safety and dual-use concerns associated with SARS-CoV-2 RBD and its engagement with receptors beyond DCs, we clarify that we are not considering RBD-containing candidates for preclinical-to-clinical advancement at this stage. As reflected in the revised Discussion, the focus of this work is to introduce a DNA-vaccine targeting concept; moving forward, we will prioritize RBD-free DC-targeting modules (e.g., anti-DEC-205 scFv, C-type lectin ligands) and evaluate protease-cleavable safety linkers to minimize off-target interactions before undertaking the fuller translational package. As a priority for follow-up, we will (i) quantify serum IL-6 and TNF-α at acute time points to exclude a cytokine surge, (ii) conduct dose–response and recovery cohorts with predefined criteria, and (iii) evaluate EV biodistribution using DiR-labeled EVs with IVIS imaging and organ ROI quantification to determine how EV modifications influence tissue targeting.

Comment 6: Discussion: Please add a paragraph on dual-use/biosafety considerations when deploying SARS-CoV-2 RBD sequences in research constructs and outline regulatory implications for clinical translation. Also, it is important to consider proposing RBD-free DC targeting alternatives (e.g., anti-DEC-205 scFv, C-type lectin ligands) and protease-cleavable safety linkers to minimize off-target interactions, some of which you already note.

Response 6:

We appreciate this important suggestion. In the revised Discussion, we have added a paragraph acknowledging the dual-use and biosafety concerns inherent to deploying SARS-CoV-2 RBD sequences in research constructs and clarify that, given these potential safety issues, we are not considering the biosafety and regulatory aspects of advancing RBD-containing candidates into preclinical or clinical development at this stage. Instead, the present study focuses on proposing a new DNA-vaccine targeting strategy; accordingly, we explicitly outline RBD-free dendritic-cell-targeting alternatives (e.g., anti-DEC-205 scFv and C-type lectin ligands) together with protease-cleavable safety linkers to minimize off-target interactions, and note that we will continue to screen and validate safer, effective DC-binding domains in future work. The revised text can be found in the Discussion (marked with tracked changes, page 20, line 694).

Comment 7: The plagiarism percentage is quite high (29%), so please reduce it.

Response 7:

Thank you for pointing out the high similarity score. We take this seriously and have conducted a thorough revision to reduce overlap and improve the clarity and originality of the manuscript.

Round 2

Reviewer 2 Report

Comments and Suggestions for Authors

All comments have accurately been addressed. Thanks.